# Importance of Surface Topography in Both Biological Activity and Catalysis of Nanomaterials: Can Catalysis by Design Guide Safe by Design?

**DOI:** 10.3390/ijms22158347

**Published:** 2021-08-03

**Authors:** Mary Gulumian, Charlene Andraos, Antreas Afantitis, Tomasz Puzyn, Neil J. Coville

**Affiliations:** 1National Health Laboratory Service, National Institute for Occupational Health, Johannesburg 2001, South Africa; maryg@nioh.ac.za; 2Haematology and Molecular Medicine Department, Faculty of Health Sciences, University of the Witwatersrand, Johannesburg 2000, South Africa; 3Unit for Environmental Sciences and Management, Water Research Group, Faculty of Natural and Agricultural Sciences, North West University, Potchefstroom 2531, South Africa; 4Nanoinformatics Department, NovaMechanics Limited, Nicosia 1065, Cyprus; afantitis@novamechanics.com; 5QSAR (Quantitative Structure-Activity Relationship) Laboratory Limited, Aleja Grunwaldzka 190/102, 80-266 Gdansk, Poland; t.puzyn@qsarlab.com; 6Laboratory of Environmental Chemometrics, Faculty of Chemistry, University of Gdansk, Wita Stwosza 63, 80-308 Gdansk, Poland; 7DSI-NRF Centre of Excellence in Strong Materials, School of Chemistry, University of the Witwatersrand, Johannesburg 2050, South Africa; neil.coville@wits.ac.za; 8Molecular Sciences Institute, School of Chemistry, University of the Witwatersrand, Johannesburg 2050, South Africa

**Keywords:** nanomaterials, nanotopography, metal and metal oxide NPs, carbon-based nanomaterials, catalysis, biological activity, descriptive toxicology, predictive toxicology, safe-by-design

## Abstract

It is acknowledged that the physicochemical properties of nanomaterials (NMs) have an impact on their toxicity and, eventually, their pathogenicity. These properties may include the NMs’ surface chemical composition, size, shape, surface charge, surface area, and surface coating with ligands (which can carry different functional groups as well as proteins). Nanotopography, defined as the specific surface features at the nanoscopic scale, is not widely acknowledged as an important physicochemical property. It is known that the size and shape of NMs determine their nanotopography which, in turn, determines their surface area and their active sites. Nanotopography may also influence the extent of dissolution of NMs and their ability to adsorb atoms and molecules such as proteins. Consequently, the surface atoms (due to their nanotopography) can influence the orientation of proteins as well as their denaturation. However, although it is of great importance, the role of surface topography (nanotopography) in nanotoxicity is not much considered. Many of the issues that relate to nanotopography have much in common with the fundamental principles underlying classic catalysis. Although these were developed over many decades, there have been recent important and remarkable improvements in the development and study of catalysts. These have been brought about by new techniques that have allowed for study at the nanoscopic scale. Furthermore, the issue of quantum confinement by nanosized particles is now seen as an important issue in studying nanoparticles (NPs). In catalysis, the manipulation of a surface to create active surface sites that enhance interactions with external molecules and atoms has much in common with the interaction of NP surfaces with proteins, viruses, and bacteria with the same active surface sites of NMs. By reviewing the role that surface nanotopography plays in defining many of the NMs’ surface properties, it reveals the need for its consideration as an important physicochemical property in descriptive and predictive toxicology. Through the manipulation of surface topography, and by using principles developed in catalysis, it may also be possible to make safe-by-design NMs with a reduction of the surface properties which contribute to their toxicity.

## 1. Introduction

Nanomaterials (NMs) are materials that have at least one dimension <100 nm [1]. They typically are made from metals, metal oxides, and carbon-based materials. This diversity of their bulk chemical compositions leads to corresponding surfaces with diverse chemical compositions. Of importance is the arrangement of the surface atoms in these NMs, which are influenced by the bulk atoms and which lead to residual valence forces of the surface atoms. As a result, the final arrangement of the surface atoms will determine their surface characteristics and the properties that will eventually determine their activity. The creation of surfaces with different properties is therefore central to the study of NMs (as it is in catalysis) and impacts, in a practical way, their different industrial and biological applications. An understanding of NM surfaces should therefore also lead to an understanding of their toxicity and, eventually, their pathogenicity.

At the nanoscale, the features considered to be responsible for the toxicity of nanoparticles (NPs) differ from those implemented for traditional chemical toxicity. The surface properties which have an impact at the nanoscale include particle size distribution, surface area, porosity, chemical composition, purity, and surface atom arrangement. These, in turn, influence the surface chemistry, coverage by atoms/molecules (involving both physical and chemical bonding), and the particle surface topography. The small size of NPs also leads to quantum confinement effects, thereby altering macroscopic properties of a material such as colour. The electronic properties of the surface, such as charge, are also affected by the small size of NPs and can affect particle aggregation, all of which have an impact on the dissolution and surface reactivity of NMs. It is, however, important to acknowledge that the dry-state characterization of these properties of NMs may be different from the properties of NMs in a wet environment (e.g., in water). This could limit our understanding of NP–cell interactions when assessing cellular uptake and toxicity. Ideally, NP surface properties should be investigated in both dry states and in typical biological surroundings.

## 2. Surface Topography of Nanomaterials

All NMs share a common feature of a large surface-to-volume ratio where surface properties determine their chemical and physical properties. Furthermore, the chemical composition and three-dimensional structure of the surface influences the nature of the chemical groups on the surface of NMs, producing the variability observed in the surface reactivity between NMs [2,3,4]. 

### 2.1. Metal and Metal Oxide NPs

For noble metals, the most commonly observed shapes are the octahedron, truncated octahedron, and cubo-octahedron, as well as the cube shape where the same metal may have different shapes, as seen in Figure 1.

The shape can be further modified by different degrees of truncation at the corners and edges of low-index facets such as (111), (110), and (100). An example of these facets for copper(II) oxide (CuO) is presented in Figure 2. These various facets can occur in different proportions [6]. It has been shown that many of the important interfacial properties of NPs depend on which of their surface facets are exposed on the particle morphology [7].

The surface will reflect the packing arrangement of the atoms in the bulk material. For example, metals with a face-centred cubic (fcc) structure will have (100) facets and would, at the surface, have four vacant bonding sites. In contrast, the (110) surfaces will have five vacant bonding sites, and a material with a (111) surface will have three vacant chemical bonds. Each facet will also have a characteristic surface energy, whose value depends on how many bonding sites there are in the surface. 

The surfaces of NMs may also consist of terraces, steps, adatoms, kinks, etc. on the surface, thereby creating a rough surface which is heterogeneous in nature (Figure 3).

Atoms, which create terraces, coordinate with many neighbouring surface atoms and thus can be viewed as having a number of free coordination sites (i.e., free valencies). These free valencies will generally be perpendicular to the surface. On the other hand, a terrace adatom (Figure 3) provides the other extreme in that it contains the fewest nearest neighbours compared to all the different surface sites. This results in it being highly coordinatively unsaturated, and it is therefore a highly active site available for reactions with external atoms or molecules. Surface atoms having a lower coordination number (CN) compared to the lattice are also considered to be catalytically and chemically more active [10,11]. Thus, the different types of surface sites will react differently with external atoms and molecules due to the difference in the number of their coordinating neighbours. The reduction of surface free energy is the driving force in minimizing particle energy. This reduction of surface free energy can occur by the processes of relaxation and reconstruction. This results in a distortion of the surface so that the bonding between surface atoms is modified. The surface atoms can also react with external atoms and molecules, and provides the basis in understanding the driving force behind catalytic reactions.

The proportions of surface atoms at corners, edges, and planes affects the outcome of possible surface reactions. These different surface atoms have different chemical environments compared with atoms in the bulk, and this may also lead to charge redistribution changes at the interface [12,13]. The structure of the surface has a strong impact on the physicochemical characteristics of the NP such as surface polarizability, surface charge [14], acidity [15], and isoelectric point [16]. In solutions, the interaction of surface atoms with external atoms and molecules leads to the generation of its hydration shell. A theoretical study concerning the influence of the size of NPs on their hydrophobicity/hydrophilicity has been explored by molecular dynamics simulations [14]. These effects have an impact on the solvation free energy of NPs. It was found that while small NPs were hydrophobic, large NPs became hydrophilic. The size of the particle impacted the water–particle interaction energy, which in turn determined the polarity of the NPs and, of course, influenced its colloidal aggregation [14]. 

A quantum confinement effect occurs when the size of the particle is small relative to the wavelength of the electron (Figure 4). A decrease in the confinement width occurs with a decrease in particle size, which in turn gives rise to a corresponding blue shift in the emission wavelength due to the increased energy bandgap (*E_g_*) [17]. The energy bandgap is related to the ability of electrons to move between orbitals in a particle (i.e., between the Highest Occupied Molecular Orbital (HOMO) of the valence band and the Lowest Unoccupied Molecular Orbital (LUMO) of the conduction band). Figure 4 shows the effects of varying particle size from 2 to 6 nm on this HOMO–LUMO gap or energy bandgap. In an atom, the ability to move an electron from the HOMO to LUMO requires a large amount of energy (since the electron is ‘confined’). However, as more atoms are added to a structure, the HOMO–LUMO gap becomes smaller and smaller and eventually leads to more facile electron transfer processes. This movement of electrons between particle orbitals will affect the wavelength and thus the colour of a particle (Figure 4).

Size, shape/geometry, and spatial organisation are the physicochemical properties that are determined by the relative number of atoms in different surface sites and, in turn, determine the surface nanotopography. Therefore, the surfaces of NMs may be adequately described by physicochemical descriptors such as surface curvature, surface energy, charge, and topography. 

### 2.2. Carbon Nanostructures

Carbon has the ability to form sp^2^ and sp^3^ bonds to other carbon atoms in order to build up a myriad of nano structures with varying sizes and shapes. These include carbon nanotubes (CNTs), carbon spheres, hollow carbon spheres, carbon nano-horns, fullerenes, carbon dots (CDs), carbon nano-onions (CNOs), amorphous carbon, and carbon nano-diamonds. Clearly, all of these will have surfaces that interact with external atoms and molecules and this will be related to their nano toxicological properties. The two extreme arrangements of carbon to form nanostructures are highlighted by CNOs and CDs (Figure 5). Here, it is seen that the carbon atoms can be arranged in layers (CDs), as found in graphite or in fragments, that create a spherical like structure such as that seen in fullerenes (CNOs). The carbon materials such as those shown in Figure 5 have surfaces with functional groups that can readily interact with atoms and molecules including proteins.

## 3. Surface Nanotopography of NMs in Relation to Their Biological Activities

Much attention has been paid to the effects of substrate surface nanotopography and on the development of new medical devices and biomaterials, especially in relation to eukaryotic cell guidance, morphology, proliferation, function, stem cell differentiation, and tissue integration [19,20,21,22,23]. Nanotopography has also been studied in relation to the attenuation of acute inflammatory response, the activation of the immune–complement system [24], controlling cancer cell function [25], and platelet adhesion and activation [26,27,28,29]. For example, in the study by Hulander et al. [29], different surfaces were prepared by adsorbing gold NPs (AuNPs) with varied inter-particle distances on smooth gold substrates. These surfaces were then pre-adsorbed with human fibrinogen and the effect of surface nanotopography on platelet adhesion and activation was studied. The authors found that while surfaces with 23% AuNP coverage resulted in a peak in platelet adhesion, the highest number of activated platelets was found on smooth surfaces with 0% coverage. The importance of nanotopography has also been considered in designing biomaterials for tissue regeneration [30]. A relatively limited number of studies have, however, reported on nanotopography in relation to the different biological effects of NMs. These may include their interaction with ligands and proteins, cellular uptake and toxicity, dissolution, bactericidal activity, and drug delivery. 

### 3.1. Interaction with Ligands

During the synthesis of NMs, the surface atoms are often covered by ligand monolayers [31,32]. The shells made of ligands can influence NMs’ growth [33] and physical properties [34], as well as mediate interactions among these surface bound ligands [35] to produce colloidal stability [36]. 

Surface-bound ligands also form the interface between a NM and its surroundings, wherein they dictate how NMs interact with each other and their environment [37,38,39]. Therefore, surface ligands are an essential component not only in NM synthesis, processing, and application, but also in determining their intracellular uptake and toxicity. This arises since the surfaces of NMs are rarely bare and, when in solution, they can adsorb, coordinate, or bind to available ligands. 

### 3.2. Effect of Curvature, Edges, and Corners on the Adsorption of Ligands and Protection against Dissolution

Ligand coverage on NMs can be influenced by surface facets or curvature [13,40]. For example, the local surface curvature of AuNPs was found to dictate the interfacial adsorption, desorption, and exchange behaviours of thiolated ligands [41].

The effect of temperature-dependent interactions between particles was shown to be strongly affected by the density of ligand coverage, as well as by the scale of various facet dimensions. Moreover, certain facet dimensions affected ligand alignment more strongly than others [42]. Facet type may also affect ligand binding. For example, polyvinylpyrrolidone (PVP) was found to bind more strongly to (111) facets of polyhedral lead(II)sulfide (PbS) nanocrystals, compared to (100) facets [43]. This suggests that the (100) facets of the polyhedral PbS nanocrystals were likely less protected by PVP compared to the (111) facets, which would result in an increased release of Pb ions from (100) facets. The (100)-facet-dependent Pb-ion release therefore suggests that the toxicity associated with these nanocrystals was probably, in this case, (100)-facet-dependent [44].

### 3.3. Effect of Ligand Density on Intracellular Uptake

Previous sections have shown that surface nanotopography affects ligand adsorption and density, which in turn may affect the interaction of NMs with cells. For example, ligand density and the presentation of ligands on NPs affects cell targeting efficacy as well as cellular uptake [45,46] and therefore affects their biological activity [47]. For this reason, surface ligand density appears to be a fundamental parameter that needs to be taken into account in order to achieve optimal interactions at the interface [48] and stabilize densely loaded ligands on NPs in nanomedicine [46].

### 3.4. Interaction with Proteins

Nanotopography has attracted much attention in relation to protein adsorption by NMs due to its importance in areas such as medical diagnostics and in the food industry [49], as well as in cellular recognition and uptake studies. The latter process is very much dependent on the manner in which the proteins are adsorbed on the surfaces of NMs [50] where surface nanotopography, in turn [51], determines protein adsorption properties such as binding affinities and surface saturation values [52]. 

### 3.5. Surface Curvature and Protein Adsorption

Many studies have indicated the importance of protein–surface interactions. The surface curvature of a NM, in particular, will affect protein adsorption and, in doing so, provide a flexibility to protein–surface interactions [53]. As a consequence, the NP surface curvature may alter the secondary structure of proteins, thereby resulting in irreversible changes to protein structure [54] and function and, ultimately, alter the bio-reactivity of the NP. The size of the NP is thus important, with smaller NPs having a higher degree of curvature (giving lower surface coverage) which will lower the protein binding affinity. In contrast, large NPs with lower surface curvatures will, for certain proteins, provide a higher binding affinity [55,56].

### 3.6. Conformational Changes of Proteins

Nanoparticle surfaces can induce conformational changes in adsorbed protein molecules. For example, it was shown that while albumin is stabilized by high surface curvature, fibrinogen is distorted by wrapping around a curved surface, thereby inducing secondary structural loss [57]. Therefore, surface curvature [58] may affect the overall bio-reactivity of the NP [59]. In some cases, this may affect the secondary structures of proteins, causing irreversible changes that may lead to structural destabilization [60] and changes in their biological properties [61]. For example, a structural change to the NP curvature upon adsorption may lead to either a loss of enzymatic activity of some proteins [62] or an increase in their toxicity [63]. It has been suggested that for nanostructured surfaces, the amount of protein is determined by its surface chemistry. For supra-molecular structures, the amount of protein depends on the topography of the surface [64,65]. AuNPs have been shown to influence conformational changes in the structure of bovine serum albumin (BSA) in a dose-dependent manner [66]. On the other hand, carbon C60 fullerene NPs did not induce any conformational changes to BSA [67], whilst zinc oxide (ZnO) NPs induced only minor conformational changes in BSA [68]. Other NPs shown to induce protein conformational changes include titanium dioxide (TiO_2_) NPs, which caused conformational changes and reduced polymerization of tubulin [69], and superparamagnetic iron oxide NPs (SPIONs), which induced an irreversible conformational change in the secondary structure of transferrin [70]. The relationship between the curvatures of different sized silica NPs (SNPs) and conformational changes to adsorbed lysozyme was also demonstrated using mesoscopic coarse-grained molecular dynamics simulations [56].

### 3.7. Nanotopography and Cellular Uptake

The intracellular uptake rate and the endocytic cellular pathway of NPs are influenced by their physicochemical properties. Interestingly, the effect of nanoscale surface topography has only recently been connected to cellular uptake pathways. For example, simulated studies showed that surface roughness decreased repulsive hydrophilic and electrostatic interactions, which promoted adhesion and cellular uptake [71]. This was also demonstrated in studies on self-assembled block copolymer NPs, where surface roughness enhanced the cell adhesion and entry into cells through reduced repulsive interactions [72]. These examples emphasize the importance of rough surfaces in leading to higher cell binding of NPs. NPs with rough surfaces have also been shown to avoid the clathrin-mediated cellular uptake pathway thus avoiding lysosomal degradation [73]. Certain studies have also shown increased gene delivery with roughened NPs [74].

### 3.8. Nanotopography and Cellular Toxicity

Recently, the relationship between the cytotoxicological effects of NMs and the presence of surface defects (including topological defects, vacancies, dislocations, grain boundaries, and surface states) has been emphasised [75,76]. For example, introducing structural defects in CNTs was shown to induce acute effects in lung toxicity [77]. The toxicity of the graphene family of NPs in relation to their surface properties has also been extensively reviewed [78]. It was reported that, at low doses, graphene oxide (GO) caused damage to the gastrointestinal tract in maternal mice. In contrast, a higher dose of GO did not show this effect. It was suggested that a low dose of GO did not show agglomeration and could therefore easily attach to the gastrointestinal surface and cause destruction through its abundant sharp edges [79].

### 3.9. Nanotopography, Dissolution and Antibacterial Activity

The dissolution of NPs is important in relation to their toxicity [80] and biodurability [81]. With metal and metal oxide NPs, face-preference is frequently encountered during dissolution [82,83]. For example, an earlier study on non-oxidative dissolution of PbS showed that (111) and (110) faces dissolved faster than (100) faces on nanocrystals. This was rationalized by the average CN of ions on each of these faces, where faces with lower CNs (namely the (110) and (111) faces) were etched more quickly than the (100) faces [84]. Conversely, it is hypothesized that octahedral shaped NPs exposing Pt(111) facets dissolve less than cubic shapes exposing Pt(100) facets in acidic environments [85]. The role of facet-dependent dissolution and bactericidal activity of NPs was also investigated. For example, silver NMs, including silver nanoplates with a (111) lattice [86,87], displayed the strongest biocidal action compared to both spherical and rod-shaped NPs, as well as Ag+ (in the form of AgNO_3_). It was therefore proposed that the high-atom-density (111) facets containing the highest reactive sites led to easier dissolution and a faster release of Ag+, therefore exhibiting a stronger antibacterial activity [88]. The reasons as to why Ag (111) facets are more prone to dissolution could be due to differences in the solvation and arrangement of the ligands, or to an instability of the Ag_2_O layer and, therefore, a preferential formation of a suboxide layer on this type of facet [89]. It was also inferred that the (111) lattice plane of triangular NPs could make better contact with, and thus damage, bacterial cell walls [90]. Similarly, the facet-dependent antibacterial effect was demonstrated in Cu_2_O crystals, where it was shown that Cu_2_O octahedral crystals bounded by (111) facets exhibited higher activity in killing *E. coli* than cubic ones bounded by (100) facets [91]. The facet-dependent antibacterial activity of ZnO NPs has also been demonstrated [92]. For example, it was found that greater antibacterial activity could be achieved with morphologies of highly exposed (0001)-Zn terminated polar facets [93]. Preferential dissolution from the (000-1) facet of ZnO NPs was also observed [94] and was demonstrated to be due to an increase in O vacancy abundance (as a result of annealing), which was predominantly responsible for the enhanced ZnO dissolution [95].

Other NPs including magnesium oxide (MgO), TiO_2_, cerium dioxide (CeO_2_), silicon dioxide (SiO_2_), and iron oxide (Fe_3_O_4_) also exhibited antimicrobial properties, where their antimicrobial activity was attributed to their reactive corners and edges [9,96]. The antibacterial activity of graphene was also shown to result from a direct contact between bacterial cells and the ‘sharp’ edges of the graphene nanowalls, which resulted in severe membrane deformation and an efflux of cytoplasmic material [97,98]. Further investigations showed that the density of graphene edges is of paramount importance in determining the effectiveness of the surface of bactericidal material [99]. In addition, graphene penetration of cell membranes was confirmed to occur in a dominant edge-first or corner-first mode for lung epithelial cells, keratinocytes, and macrophages [100]. 

In summary, it can be seen from previous sections that nanotopography of NMs plays an important role in determining their biological activities. This is summarized in Figure 6. In-depth knowledge of these very same surface properties is therefore critical in understanding their interactions with cells and tissues, which determine not only the potential toxicity of NMs but also their biodistribution, degradation, biocompatibility and, ultimately, their pathogenicity [101].

## 4. Surface Nanotopography of NMs in Relation to Their Catalytic Properties

There is a clear link between the surfaces of NPs and catalysis. This connection is provided by both the underlying structure below the surface and the ‘roughness’ of the surface due to surface atom unsaturation and surface defects. The catalytic activity of metal nanocrystals is highly dependent on the nature of their surface structure, where exposure of different crystallographic facets, together with the increased number of edges and corners, are key parameters [102,103]. Therefore, NPs of different shapes are highly desirable as catalysts [104,105,106,107].

Heterogeneous catalytic reactions occur on the surface of solid catalysts and involve elementary surface chemical processes such as adsorption of reactants from a reaction mixture, surface diffusion, and reaction of adsorbed species, followed by the desorption of reaction products. The acceleration of a chemical reaction is due to the high reactivity of surface atoms that facilitates bond breaking and bond rearrangement of adsorbed molecules. One of the properties of metallic NPs is their high number of surface atoms, which increase with decreasing particle size. These surface atoms are the active sites for catalysis. Control of NP size is thus of high importance. In addition, surface atoms that are located at edges or in corners are more active than those in planes, and their number also increases with decreasing particle size. An example is that of molybdenum disulphide (MoS_2_) edge sites that perform as active centres for catalytic reactions [12]. 

At the nanoscale, both quantum chemistry and the classical laws of physics may apply. For example, the strong bonding present in NMs may lead to extensive delocalization of electrons, which in turn may vary with the particle size. Specifically, the observed enhanced catalytic properties of NPs have been linked to an increase in the undercoordinated facets, which bind reaction intermediates more strongly [108,109,110]. Under certain conditions, however, NP catalysts can be synthesized such that the shape, and consequently the surface atom orientation, is kinetically trapped into a non-equilibrium shape such as a cube, triangle, platelet, or rod [106]. These NPs with different shapes have been shown to have varied activity and selectivity as well as stability in catalytic reactions [111]. An example of the role of facets in metal-based NP catalysts is also illustrated by cubic palladium nanocrystals, with wholly exposed (100) facets exhibiting the highest activity while the rhombic dodecahedra with (110) facets shows the lowest electrocatalytic performance [112]. In addition, active (110) and (100) planes show significantly enhanced redox properties that determine their catalytic activity in CO oxidation [113,114]. The importance of the effect of the ratio of (001) and (101) facets of anatase TiO_2_ on the photocatalytic reduction of CO_2_ was also reported [115]. Moreover, anatase TiO_2_ nanosheets with a high ratio of (001)-to-(101) facets displayed the best photocatalytic activity for the reduction of CO_2_ [116]. Theoretically, the (001) facets exhibit a surface energy higher than that of the (101) facets, and appropriate exposure of the (001) facets can improve the catalytic activity of TiO_2_ NMs [117]. 

In carbon-based NMs, the controlled curvature or the orientation of the graphene layers have been shown to dictate important properties. For example, the curvature in fullerenes or CNTs is due to the introduction of pentagons and heptagons that changes the properties of a CNT relative to a graphene sheet [118]. Thus, porous carbon-based catalysts such as single walled CNTs (SWCNTs) show high catalytic activity of SO_2_ oxidation, and there is a direct correlation between the decrease in the barrier for the oxidation and the increase of curvature (size) on the SWCNTs [119].

## 5. Surface Nanoscale Topography as Effective Design Strategy

As illustrated in the previous section, nanoscale topography has long been implemented in the design of NMs for their catalytic applications. Moreover, the use of nanoscale topography in drug delivery is an emerging concept that has been proposed in a number of applications. More recently, similar limited approaches have also been proposed for the design of safe NMs.

### 5.1. Nanotopography and Catalysis by Design

An overview summarising the effects of metal nanocrystals on catalytic activity in “Designing Nanoparticle Systems for Catalysis” has recently been reported [120,121]. As mentioned above, the key factors that need to be taken into account in the modelling of heterogeneous catalysts are the surface structure, facets, edges, and corners [104,105,106,122]. Theoretical studies on adsorption energies of reaction intermediates [123,124] and activation energies of transition states have already been reported [125]. In these studies, the dynamic character of a catalyst is required, since the identification of any reaction intermediates may be responsible for surface atom transformations [126,127]. Theoretical studies on the catalytic and structural properties of NMs, in addition to experimental studies, are also needed. These include density functional theory (DFT)-based computational simulations and HOMO–LUMO energy gap calculations. This is now a widely used computational method in catalysis and was developed to rationalize experimental findings and to provide a framework for understanding surface chemical bonding [128].

DFT-based techniques are now routinely used to evaluate and interpret observations made in surface science, catalysis, and materials science. For example, a review on the fundamentals of DFT and computational methods on surface chemistry and catalytic reactions in the area of heterogeneous and electrochemical catalysis has summarised important developments in this area [129]. Thus, the approach of using DFT calculations for surface processes has led to the possibility of using DFT approaches in computer-based catalyst designs. DFT calculations can now rationalize the effect of the structural features identified on their catalytic properties [128]. Computational modelling in heterogeneous catalysis still requires the use of well-defined crystallographic planes. The use of DFT calculations, together with Boltzmann ensembles of NPs, can provide different surface morphologies under varying reaction conditions (such as temperature and gas-phase chemical potential). The calculations can also take into account the different distributions of active sites [130]. DFT-based techniques have therefore effectively transformed our understanding of the fundamentals of surface science, catalysis, and material science.

AgNPs have been well studied and their different structures have been linked to different activities. DFT simulations on the configurations of cubo-octahedral and icosahedral AgNPs was carried out, and it was shown that small decahedral structures exhibited superior catalytic activity when compared with cubo-octahedral structures [131]. Moreover, during the identification of the active sites of a Cu/ZnO/aluminium oxide (Al_2_O_3_) catalyst used in the industrial methanol synthesis reaction, DFT calculations were conducted to rationalize the effect of the structural features that have been previously identified to be relevant for their catalytic properties [132]. Most recently, the inclusion of different shapes, volume-to-surface ratios, and surface nanotopography (facets, steps, adatoms, etc.) was proposed in relation to their activity. Subsequently, the role of the topological aspects (sizes and shapes) of metallic NPs on the energetics of the processes that are related to biochemical activity was emphasized. The methods for the calculations of numerical descriptors of NPs, and the minimal model for these calculations, was discussed. Calculations were performed for a set of metals (Al, Fe, Cu, Ag, Au, and Pt), taking into account the diversity of atomic structures of real metallic NPs for different models such as (001) and (111) surfaces, nanorods, and two different cubic NPs of 0.6 and 0.3 nm size. Descriptors were proposed with consideration of the dependence of chemical activity on the size and shape of the NPs [133]. For example, CO activation on a hexagonal close-packed (hcp) structure and the metastable fcc cobalt NPs were compared using DFT calculations to confirm the dependence of the activity on their crystallographic structure and morphology [134]. Moreover, the DFT calculations showed that the preferentially exposed (101) and (020) facets might present particular active sites in Fischer–Tropsch synthesis for syngas conversion into olefin production and inhibit methane formation by manganese (Mn)-promoted cobalt carbide nanoprisms [135].

An example of the success of the DFT approach has been in the use of d-band model, which has allowed the bonding strength on transition metal surfaces to be correlated with the local electron density in the d-band of bond sites. When applied to gold, the d-band model could rationalize why bulk gold was noble while small gold clusters were not [136,137]. The approach also provided an explanation as to the enhanced activity of surface defect sites such as steps and kinks [138].

### 5.2. Nanotopography, Drug Delivery and Therapeutic Applications by Design

NMs used in different biomedical applications including imaging, diagnostics, and therapeutics, where the importance of nanotopography is emphasized as an effective strategy in determining drug loading, cell penetration, and body clearance [73]. For example, the importance of surface roughness of polyethylenimine (PEI)-modified SNPs on pDNA loading was reported for gene delivery, where the surface nanoscale topography was demonstrated to protect the DNA molecules effectively against nuclease degradation with respect to other structures [74]. Other studies also emphasized the importance of NP surface roughness as an important parameter for enhanced drug loading and cellular delivery [139,140,141]. Gene transfection efficiency has been shown to increase due to roughened NM surfaces. This is explained by the surface ‘altering the intracellular trafficking pathway, or by protecting the genetic molecules from an enzymatic degradation’ [73]. 

It therefore appears that rough NM surfaces may provide enhanced surface areas that allow more interactive sites with biomolecules and drugs, ultimately leading to increased loading capacity and cellular delivery. It can therefore be concluded that the engineering of NPs with unique surface nanoscale topography may offer an effective design strategy to the development of NPs for drug and gene delivery with enhanced therapeutic efficacy. Such an effective design strategy was demonstrated for cerium oxide NPs, where (111)/(100) nanopolyhedra with a high concentration of Ce^4+^ ions promoted catalase mimetic activity, while (100) nano/submicron cubes and (111)/(100) nanorods with high Ce^3+^ ion concentrations enhanced superoxide dismutase (SOD) mimetic activity [142]. Cadmium chalcogenide has been used in cancer treatment, but only cadmium chalcogenide nanocrystals with specific facets (i.e., (100) facet of cadmoselite and (002) facet of greenockite) were useful. These facets preferentially bound with transferrin in a complex protein matrix via an inner-sphere process. The specific facets enhanced the receptor-mediated delivery of the nanocrystals into cancer cells [143]. The above examples indicate the promise of facet engineering of crystalline NMs in biological applications.

### 5.3. Nanotopography and Nanomaterials Safe-by-Design

One of the earliest applications of nanotechnology was in catalysis [144]. As discussed in the previous section, nanocatalysis could ensure high catalytic activity and selectivity with excellent yields, traditionally related to homogeneous catalysis (as well as enhanced product separation and catalyst recovery) typically associated with heterogeneous catalysts [145,146,147]. The development of NMs growth theory has enabled the synthesis of NMs with adjustable shapes [148]. These have included nanocubes [149,150], nanooctahedra [151], nanotetrahedra, and nanoicosahedra [152]. It has therefore become evident that a correlation between structure and activity of nanocatalysts was possible. 

To avoid the toxic effects of NMs that are intended to be used for biological and industrial applications, numerous publications have proposed different strategies for a safe-by-design synthesis. In these processes, the surface of the NP is modified by functionalisation. Examples include the functionalisation of surface molecules (such as SNPs) with amino or carboxyl groups to decrease their cytotoxicity to parenchymal hepatocytes [153]. Grafting, in which functional moieties such as probe molecules and targeting ligands are covalently attached to NMs, has also been reported [154,155]. Loading is also an important consideration for safe-by-design NPs. The non-covalent loading of organic molecules, polymers, or biomolecules onto a NM surface can occur via the typical range of weak interactions such as van der Waals forces, hydrophobic and electrostatic interactions, π–π stacking, or hydrogen bonding [156,157]. Typical samples that have been coated and encapsulated using this approach are amphipathic lipids, polymers, silica, zinc sulfide (ZnS), and biocompatible materials. Using this approach has led to the prevention of the release of toxic components from a NM core. This has led to a reduction in in the adverse effects of the NM [158,159]. Doping, which involves the introduction of a small percentage of foreign atoms into an otherwise pure material, has also been implemented to design less toxic NMs [160].

Safe-by-design strategies may be supported by nanoinformatics modelling, which reduces the number of experiments and assists in selecting the most optimal conditions. The main idea behind this type of modelling is to describe mathematically quantitative relationships between the structural variations of NPs in a group and their activity. For example, Mikolajczyk et al. [161] developed models for predicting both photocatalytic activity and toxic effects against Chinese hamster ovary cells for twenty nine TiO_2_ NMs with clusters of noble metals (i.e., Ag/Au/Pd/Pt). In addition, safe by design nanoinformatics models are available through Enalos, NanoCommons, and NanoSolveIT Cloud Platforms as web services with user-friendly graphical interfaces. For example, the ‘Enalos Nanoinformatics Cloud Platform: A Safe-by-Design Tool for Functionalised Nanomaterials’ is an online tool for the design of MWCNTs with desired toxicities and biological activities [162]. In addition, the MS^3^bD (MScubed: Molecular, Size and Surface based Safe by Design) model [163] may be used for the design of NMs with desired ζ-potential properties assisted by a read-across concept.

Surprisingly, both experimental and computational studies conducted so far have not yet considered the issues of nanotopography in the safe-by-design synthesis of NMs. Recently however, the importance of nanotopography was acknowledged where the passivation of nanotopographically active defect sites was emphasized [164,165]. One example is the facet dependent dissolution of Pb, as previously discussed, which was modified by facet dependent coverage by organic molecules such as PVP. Low Pb dissolution from (111) facets was shown to be due to the strong protection afforded by the addition of PVP. However, the uncovered (100)-facets led to Pb dissolution. Based on this facet-toxicity relationship, a safe-by-design strategy was used to prevent Pb dissolution from (100) facets. This was achieved through the formation of atomically thin Pb-chloride adlayers, which resulted in safer-to-use polyhedral PbS nanocrystals [43].

The structures of NPs can change dynamically under reaction conditions [166]. As the size of NPs decreases, their properties change dramatically where the surface atoms are undercoordinated and therefore are less stable [167]. The distribution of surface sites (e.g., terrace, steps, edges, corners, etc.) also changes with particle size [168]. For example, the fraction of terrace atoms decreased continuously when decreasing the particle size while the atoms at edges and corners, which have fewer direct neighbours than terrace atoms, gradually increased and became dominant when the size was reduced down to 2 nm [169]. This meant an increase of corners and edges that led to a higher degree of surface unsaturation, since the CNs of atoms at corners and edges are both lower than that on a terrace. Therefore, the dynamic evolution of active sites has always been an essential topic of catalysis and it has become clear that maximization of surface, edge, and corner sites should be the criteria for designing superior nanocatalysts. 

This raises optimism that using a similar approach via the control of nanotopography will possibly lead to the synthesis of NPs with reduced toxicity. The information on nanotopography could be further used as descriptors in nanoinformatic-based studies for developing models for predicting toxicity. In effect, the production of tailor-made safe-by-design NMs will be possible through precise manipulation of their nanotopography.

## 6. Conclusions

NM surfaces are involved in many important chemical and biological processes in nature and in various industries, largely because of their high chemical reactivity. The role of nanotopography in both catalysis and biological activity of NMs is summarised in Figure 7. 

An understanding of surface processes at the molecular level is not trivial, since this understanding requires a knowledge of the surface at the atomic level. This requires techniques that can ‘see’ the surface at this nanometer level. The development of surface-specific techniques has led to the development of modern surface chemistry. These techniques have been used to investigate surfaces at the required spatial, time, and chemical resolution. In particular, the ability to use atomic force microscopy, various electron microscopies, synchrotrons, etc. has become crucial in the development of surface science principles. These techniques have been extensively applied in evaluating size and shape-controlled NP synthesis, nanostructure assembly, and material property characterization [31,170]. The use of experimental techniques, combined with theoretical simulations [171], has also been important for the rational design of surface properties for technological applications [172]. Advanced applications of NMs in medicine and concerns relating to their health effects may raise hope for implementing similar approaches combining experimental techniques with theoretical simulations in order to achieve in-tuning surface structures and produce safe-by-design NMs. 

## Figures and Tables

**Figure 1 ijms-22-08347-f001:**
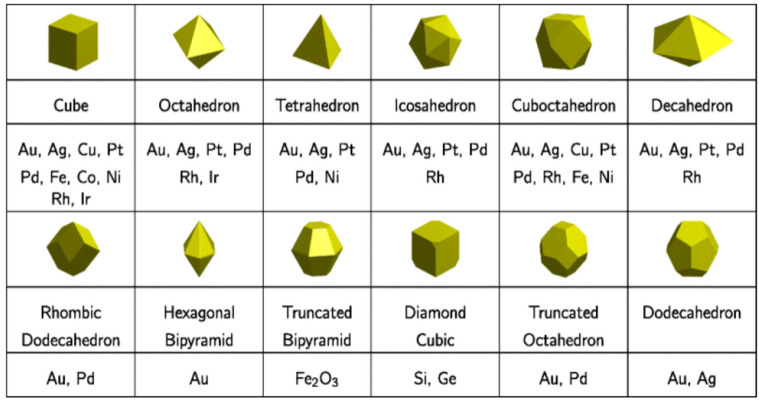
Shape-dependent synthesized nanoclusters. Reprinted with permission from Reference [5]. Copyright 2020 Springer Nature.

**Figure 2 ijms-22-08347-f002:**
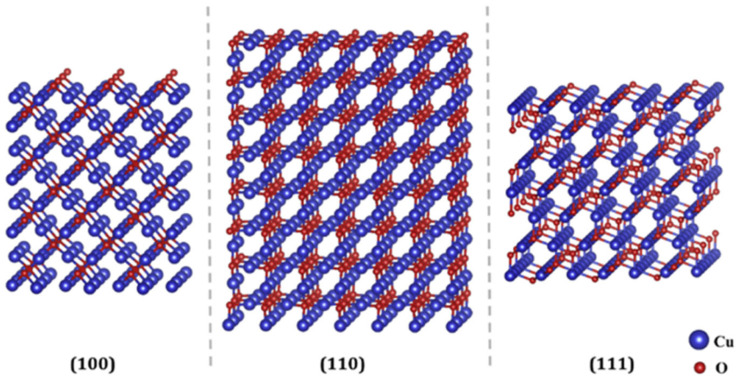
Side view scheme of the selected (100), (110), and (111) slabs of a copper(II) oxide (CuO) cube. Adapted with permission from Reference [8]. Copyright 2021 Elsevier.

**Figure 3 ijms-22-08347-f003:**
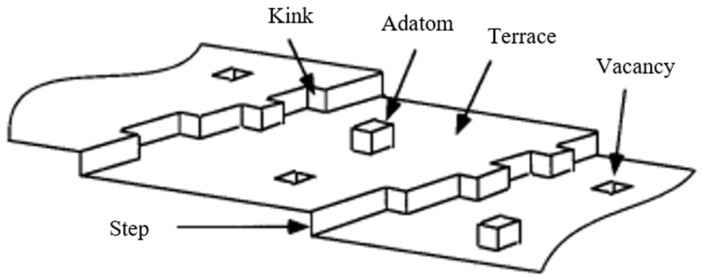
Simple block model of defects on a single-crystal surface. Reprinted with permission from Reference [9]. Copyright 2021 American Chemical Society.

**Figure 4 ijms-22-08347-f004:**
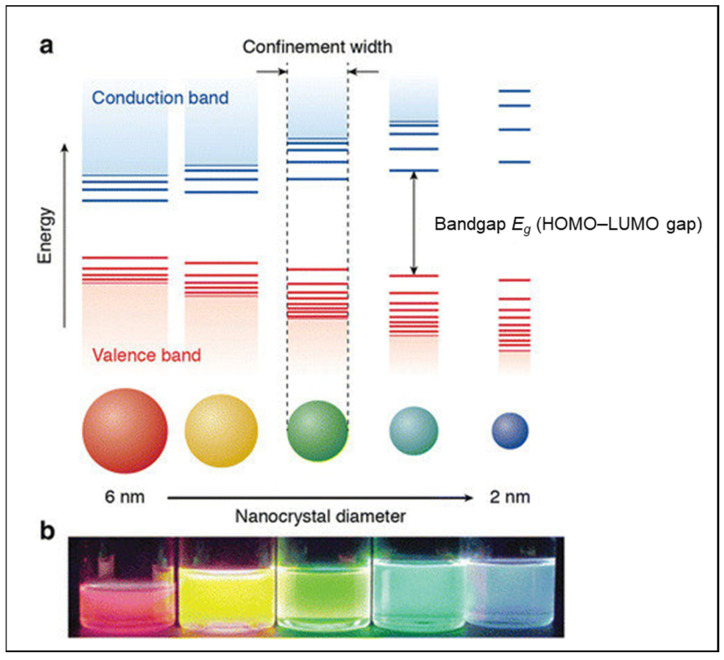
A Schematic representation of the quantum confinement effect: (**a**) The bandgap (or HOMO–LUMO gap) of the semiconductor nanocrystal increases with decreasing size, while discrete energy levels arise at the band-edges; (**b**) Five colloidal dispersions of cadmium–selenium quantum dots (CdSe QDs) under UV excitation ranging from 6 nm (red) to 2 nm (blue). Adapted with permission from refs. [17,18]. Copyright 2021 Springer Link (http://creativecommons.org/licenses/by/4.0/; accessed on 7 June 2021).

**Figure 5 ijms-22-08347-f005:**
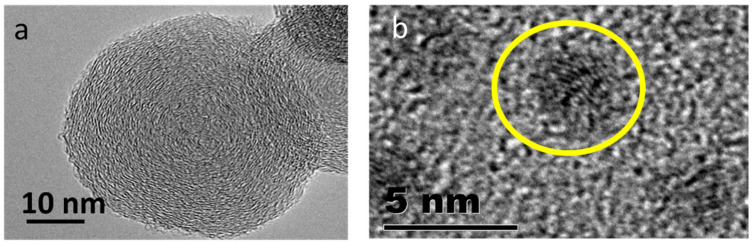
TEM image of a carbon nano-onion (**a**) and carbon dot (**b**). Image courtesy from Neil J. Coville.

**Figure 6 ijms-22-08347-f006:**
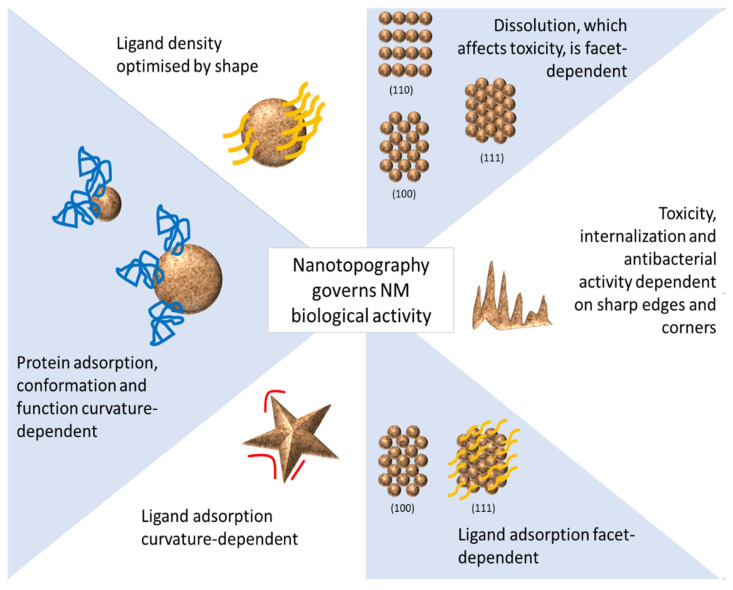
Nanotopography and biological activities of NMs.

**Figure 7 ijms-22-08347-f007:**
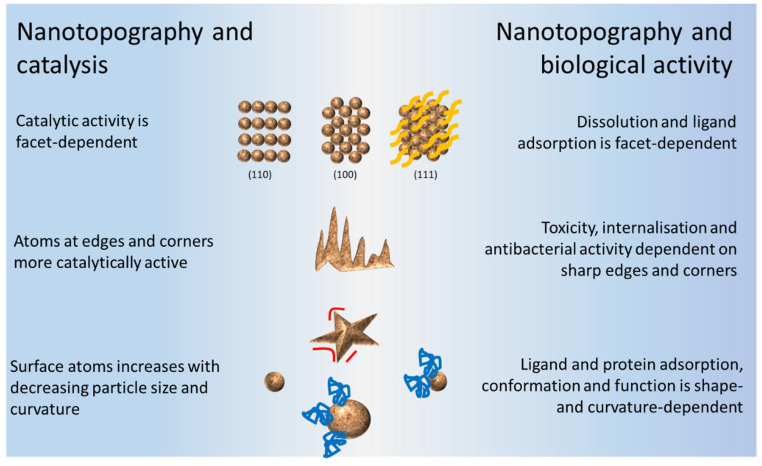
The role of nanotopography in both the catalytic and biological activities of NMs.

## Data Availability

Not applicable.

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
