# Peer review of "Importance of Surface Topography in Both Biological Activity and Catalysis of Nanomaterials: Can Catalysis by Design Guide Safe by Design?"

_ijms, 2021, doi:10.3390/ijms22158347_

Round 1

Reviewer 1 Report

Comments on manuscript, ijms-1217880 :

Title: “Importance of Surface Topography in both Biological Activity and catalysis of Nanomaterials: Can Catalysis by Design guide safety by design?”

In this manuscript, the authors tried to review the role of Surface Topography on both Biological Activity and Catalysis of Nanomaterials.  I am sure this topic can be very important and interesting for International J. of Molecular Sciences, but I am afraid this topic is too much broad to be covered deeply and meaningfully in one review manuscript.  I’d like to recommend the authors to rewrite the manuscript and submit it again, if they want to publish this manuscript in International J. of Molecular Sciences.  Some comments on this manuscript are as follows.

  1. There are some errors in English sentences and they should be corrected.
  2. Some contents in this manuscript are the topics in relevant textbooks and are too basic and not so much informative for a review paper. The authors should modify the contents to be appropriate for a review paper.
  3. Despite the importance of review topic, this review manuscript provides some fragmentary information, not comprehensive related with this topic. The information here is nothing wrong, but not so much informative. The authors can write this manuscript in more systematic way with more important and concrete information.
  4. The contents in ‘SURFACE NANOTOPOGRAPHY OF NANOMATERIALS IN RELATION TO THEIR BIOLOGICAL ACTIVITIES‘ and ‘SURFACE NANOTOPOGRAPHY OF NPS IN RELATION TO THEIR CATALYTIC PROPERTIES ‘ look separate and look in different papers. The contents should be integrated to be in one manuscript. 

Reviewer 2 Report

The review puts together the relevance of the nanotopography of the surface of nanomaterials with both their catalytical and biological activities, including consideration of safe-by-design concepts for their control.

The manuscript is generally well-written and structured, there are, however, several more or less critical points where revision is necessary:

  • Figure 1, caption: the images shown are not ‘synthesis’ but ‘synthesized clusters’
  • Figure 2: explain the ‘c’ and ‘b’ arrows, and also show ‘a’.
  • Figure 3: poor quality
  • Lines 120-121: ‘by reducing the surface free energy by the ..’: predicate is missing.
  • Line 129: ‘CN’ can be introduced earlier.
  • Line 135: ‘The solvation free energy …becomes more favorable as the size increseas..’: It would be very good to have a practical example here with numbers (e.g. Ag NP of different sizes and their solvation free energy). Especially in a review the reader should get presented the real picture from the available literature.
  • Figure 4: The term ‘Confinement width’ is not related in the text (and also not explained); HOMO & LUMO should be also marked on the figure.
  • Figure 4, Caption: ‘The energy separation between the band-edge levels also increases…’: This is the same with the previous sentence.
  • Lines 169-170: It would be good to show exemplary a CNO and a CD how to they look like.
  • Line 174: ‘…however, and are produced…’: unclear sentence.
  • Line 196: the sentence with ‘platelet adhesion’ is too scarce. Please explain that the platelet as special particle SHAPE plays a defining role in adhesion. Again, example of type of platelet material from the cited literature would be welcome.
  • Lines 197-201: Long phrase without predicate.
  • Line 211: unclear sentence: ligands mediates interactions between ligands?
  • Lines 274-275: unclear sentence.
  • Lines 290-291: unclear sentence with the toe ‘that’s.
  • Line 283 (also line 291): ‘AuNPsNPs’. Why NPs twice? With explain or correct.
  • 6: ‘100 facets favors dissolution’: i) one the previous page is stated that the densely packed facets favor dissolution, ii) e’100 facets favor dissolution’.
  • 6: the light green ligands are invisible on the light blue background, please correct.
  • Page 17, first paragraph: poor English. Please revise the language.
  • Page 17, 3rd paragraph: sounds like Conclusions.
  • 7: The ‘relationship’ is not at all explained. Please explain the arrows and terms better, possibly redraw the diagram.
  • Line 647: ‘that’ is obsolete.
  • 8: the ligands are invisible, please change the color.
  • 8: ‘Nanotopography’ is written twice false, please correct.
  • **************************************************************

Round 2

Reviewer 2 Report

All points have been revised carefully. I have no further comments and reccomend the acceptance of the manuscript.